# Harnessing the Power of Stem Cell Models to Study Shared Genetic Variants in Congenital Heart Diseases and Neurodevelopmental Disorders

**DOI:** 10.3390/cells11030460

**Published:** 2022-01-28

**Authors:** Xuyao Chang, Mingxia Gu, Jason Tchieu

**Affiliations:** 1Division of Developmental Biology, Cincinnati Children’s Hospital Medical Center, Cincinnati, OH 45229, USA; xuyao.chang@cchmc.org (X.C.); mingxia.gu@cchmc.org (M.G.); 2Molecular and Developmental Biology Graduate Program, University of Cincinnati College of Medicine, Cincinnati, OH 45267, USA; 3Center for Stem Cell and Organoid Medicine (CuSTOM), Cincinnati Children’s Hospital Medical Center, Cincinnati, OH 45229, USA; 4Division of Pulmonary Biology, Cincinnati Children’s Hospital Medical Center, Cincinnati, OH 45229, USA; 5Division of Molecular Cardiovascular Biology, Cincinnati Children’s Hospital Medical Center, Cincinnati, OH 45229, USA; 6Department of Pediatrics, University of Cincinnati College of Medicine, Cincinnati, OH 45267, USA

**Keywords:** pluripotent stem cells, differentiation

## Abstract

Advances in human pluripotent stem cell (hPSC) technology allow one to deconstruct the human body into specific disease-relevant cell types or create functional units representing various organs. hPSC-based models present a unique opportunity for the study of co-occurring disorders where “cause and effect” can be addressed. Poor neurodevelopmental outcomes have been reported in children with congenital heart diseases (CHD). Intuitively, abnormal cardiac function or surgical intervention may stunt the developing brain, leading to neurodevelopmental disorders (NDD). However, recent work has uncovered several genetic variants within genes associated with the development of both the heart and brain that could also explain this co-occurrence. Given the scalability of hPSCs, straightforward genetic modification, and established differentiation strategies, it is now possible to investigate both CHD and NDD as independent events. We will first overview the potential for shared genetics in both heart and brain development. We will then summarize methods to differentiate both cardiac & neural cells and organoids from hPSCs that represent the developmental process of the heart and forebrain. Finally, we will highlight strategies to rapidly screen several genetic variants together to uncover potential phenotypes and how therapeutic advances could be achieved by hPSC-based models.

## 1. Introduction

Congenital heart diseases (CHD) are known as the most common congenital defects affecting around 1% of births in the US [1], ranging from mild ones such as atrial septal defects to anatomically complex ones such as hypoplastic left heart syndrome (HLHS). While recent advances in surgical procedures and clinical care management have dramatically improved patients’ cardiac function, addressing noncardiac co-morbidities has become increasingly important to the enhance quality of life of patients [2]. Of note, neurodevelopmental impairments have become the major long-term co-morbidities in CHD populations, with an incidence rate of up to 50% in children requiring cardiac surgeries [2]. For example, children with CHD are more likely to develop autism spectrum disorder [3], a neurodevelopmental disorder (NDD) characterized by social interaction deficits and restricted & repetitive behavior. While traditionally intraoperative factors have been regarded as the major contributing factor to NDD in CHD, recent research has suggested an opposite direction that preoperative and innate patient factors are playing more important roles [2]. This is also in accordance with the findings that in CHD patients, changes in fetal brain size and abnormalities in brain structure have been detected as early as the second trimester [4]. Overall, this suggests that neurodevelopment is disturbed in CHD but the underlying mechanisms are less understood.

Although heart and brain development occur in spatially distinct regions, they share similar genetic factors and signaling pathways that are critical for an accurate developmental trajectory. Thus, one would posit that genes with pathogenic variants expressed in both heart and brain development can lead to the co-occurrence of CHD and NDD. In line with this, recent studies using whole-exome sequencing of patients with CHD have revealed a significant overlap between high-risk CHD and NDD genes [5,6,7]. However, the impact of shared genetic variants on both heart and brain development is still unknown. This could be due to both the unawareness of the shared genetic link and the insufficiency of current animal models to recapitulate patient phenotypes. Traditional animal studies are instrumental in delineating key molecular events during development, but species-specific features can interfere with determining human-related conclusions.

The differentiation of human pluripotent stem cells (hPSCs) has drawn significant attention in recent years, as these cells provide researchers with a simple, readily available, and highly relevant model to study complex human diseases such as CHD and NDD. These methods can avoid some limitations of animal models and provide insights into a time of development that is not easily observed in humans. Importantly, the differentiation of hPSCs is modular and can be based on the complexity desired. One can generate a single disease-relevant cell type in a two-dimensional (2D) format for cell autonomous studies or perform phenotypic screening, and if multiple cell types are required to complete a biological circuit or process, one can combine cell types together to form a minimal model. To observe the development of an organ, hPSCs can be directed to differentiate in a three-dimensional (3D) format to increase the complexity and gain insights into cell-autonomous and cell-non-autonomous effects. The potential of this model is further fueled by the development of efficient genome editing tools and induced pluripotent stem cell (iPSC) technology for modeling patient-specific genetic variants. Therefore, hPSC-based models can investigate whether genes with shared genetic variants can induce CHD and/or NDD to further our understanding of these co-occurring disorders.

In this review, we will first provide a brief overview of heart and brain development in order to pinpoint important developmental concepts that will be mentioned after. Secondly, we will discuss shared genetics in CHD and NDD by analyzing published patient-specific genetic variant data and focusing on important shared pathways. We will then talk about difficulties in modeling co-occurring disorders and summarize current methods to derive mesodermal cells, including cardiomyocytes and endocardial cells and ectodermal cells including cortical neural progenitors, neurons, and glia from hPSCs. The advantages and limitations of hPSC-based models have been reviewed elsewhere [8,9,10] and will not be extensively covered in this review. Finally, we will highlight screening strategies for identifying phenotypes caused by shared genetic variants and potential therapeutic options provided by hPSC-based models.

## 2. Brief Overview of Heart and Brain Development

### 2.1. Heart Development

The heart is primarily derived from the mesoderm as cardiac-fated cells will first migrate towards the anterior end of the embryo to form two major progenitor populations: the cells of the first (FHF) and second heart field (SHF). Cells from FHF will form the early linear heart tube and future left ventricle, while cells of the SHF will migrate towards the heart tube and contribute to the formation of additional structures such as the right ventricle, both atria and the outflow tract (OFT) [11]. The heart tube will further undergo cardiac looping to establish right-left asymmetry. At this time, two layers of cells are formed: the inner endocardial layer and the outer myocardial layer. The myocardial layer will give rise to the future myocardium with chamber-specific properties and differentiate into the inner trabeculated layer and outer compact layer. Endocardial cells will undergo endothelial-mesenchymal transition (EndoMT) to invade the region between the endocardium and myocardium termed the cardiac jelly to promote swelling and give rise to the endocardial cushion that forms the valve and septum. The epicardium is formed by a distinct progenitor population called proepicardium to cover the entire embryonic heart. These cells will also give rise to the mesenchymal cells in the heart and contribute to the formation of coronary vessels. 

### 2.2. Brain Development

Brain development begins from the specification of neuroectoderm. The most anterior region forms the forebrain, and the midbrain, hindbrain, and spinal cord are derived from more posterior regions. Signaling molecules secreted from the overlying epidermis and the underlying notochord will also establish the dorsal-ventral polarity of the neural tube [12]. The dorsal forebrain region is the precursor for cortical excitatory neurons, which constitute around 80% of neurons in cerebral cortex [13]. The ventral forebrain will form the ganglionic eminences and contribute to the formation of several regions of the brain including cortical inhibitory interneurons that can migrate tangentially towards the dorsal forebrain. 

Embryonic neurogenesis in the dorsal forebrain starts from neuroepithelial cells becoming radial glial cells (RGC) and undergoing symmetric and asymmetric divisions to expand the progenitor pool or form the six cortical layers in a sequential “inside-out” fashion, respectively. The cell bodies of early born neurons will be located in the deep layers (Layers 5/6) and later born neurons will settle in the superficial layers (Layers 2/3/4). Glutamatergic excitatory neurons can be generated either directly from RGCs or indirectly from secondary basal progenitors that are differentiated from RGCs, such as intermediate progenitor cells (IPC) in the subventricular zone (SVZ) or outer radial glial cells (oRGC) in the outer subventricular zone (oSVZ). Interestingly, the oSVZ is primate-specific and is thought to give rise to the massive expansion of the cortex. After neurogenesis tapers, RGCs will switch to a gliogenenic program to produce macroglia such as astrocytes and oligodendrocytes. Astrocytes are important for the regulation of brain homeostasis, while oligodendrocytes will form the myelin sheath that covers and protects the axon bundles. 

## 3. Shared Genetics in CHD and NDD

Development of the heart and brain diverge early; however, these spatially separated developmental programs are not mutually exclusive and share common developmental signaling pathways and epigenetic transitions to induce cell fate transitions. In order to dissect shared genetics in CHD and NDD, we performed gene ontology analysis to identify altered pathways that are shared in both diseases using published genetic variant data found in patients diagnosed with either CHD or NDD. 

Two types of variants were detected in patient whole-exome data: inherited and de novo mutations (DNMs). DNMs arise during gamete formation and are considered more harmful than inherited mutations as they do not undergo natural selection. Therefore, genes with predicted damaging DNMs from clinical cohorts of CHD [6] and NDD [5] (referred as CHD genes and NDD genes, respectively) are analyzed by gene ontology (GO) analysis to identify unique and shared significant GO terms (FDR < 0.05) (Figure 1A and Appendix A, Appendix A). We identified GO terms that belong to either CHD genes or NDD genes (Appendix A). Terms associated with cardiac function such as “cardiac conduction (GO:0061337)” and “cardiac muscle contraction (GO:0060048)” are only found in CHD genes, while most GO terms associated with neuronal identity and function such as “postsynaptic density assembly (GO:0097107)” and “regulation of synapse structure or activity (GO:0050803)” are only found in NDD genes. Among shared GO terms between CHD genes and NDD genes, “heart development (GO:0007507)” and “brain development (GO:0007420)” highlight similar genetic factors important for the development of both organs. Furthermore, “Wnt signaling pathway (GO:0016055)” is the most significantly enriched term in both CHD genes and NDD genes compared to other important developmental pathways such as Retinoic Acid (RA), Fibroblast Growth Factor (FGF), Bone Morphogenic Protein (BMP), Transforming Growth Factor-β (TGF-β), Hedgehog (HH), and Notch signaling (Figure 1B, Appendix A). To validate this finding, genes that are shared by CHD and NDD (referred as CHD and NDD genes) are also analyzed by GO analysis (Figure 1A, Appendix A). We found that “canonical Wnt signaling pathway (GO:0060070)” is also a significantly enriched term in CHD and NDD genes (Figure 1B and Appendix A). This is in agreement with previous work demonstrating the importance of the Wnt pathway in CHD [14] and NDD [15]. While this analysis highlights Wnt signaling as the most significant, we cannot discount the contribution of other signaling pathways in these disorders. 

We next assessed whether epigenetic related factors are enriched in CHD genes or NDD genes as epigenetic changes are synonymous with cell-fate transitions during development [16]. Intriguingly, chromatin associated genes are a significant portion of shared risk genes between CHD and NDD (29% in Homsy et al. [5], 43% in Jin et al. [6]). Analysis of GO terms related with epigenetic regulation identified “histone lysine methylation (GO:0034968)” and “histone H3-K36 methylation (GO:0010452)” as significantly enriched in both CHD genes and NDD genes (Figure 1C and Appendix A). Furthermore, GO terms related with major H3 methylation (H3-K4, H3-K9, H3-36), except for H3-K37, are enriched in CHD and NDD genes. This suggests that although H3-K36 methylation might be affect more, chromatin status and function could contribute to the links between the onset of CHD and NDD. 

Based on these analyses, we will briefly discuss the role of Wnt signaling and key epigenetic factors shared in heart and brain development. These established studies will serve as an important reference for future stem cell modeling to explore further how pathogenic variants in CHD and NDD lead to aberrant heart and brain development.

### 3.1. Wnt Signaling Is Critical in Many Stages of Heart and Brain Development

Wnt signaling serves as one of the most complex signaling transduction pathways in development [17,18]. In cardiac development, balancing Wnt activity is associated with progenitor competency and is important for directing cell-fate towards the cardiac lineages (for detailed reviews [19,20]). While canonical Wnt signaling is essential for mesoderm induction, persistent activation represses the cardiac lineage due to failure to form cardiac progenitors. Noncanonical Wnt activity, however, promotes the formation of cardiac progenitors. After specification, the expansion of this progenitor pool will again require canonical Wnt signaling, and the terminal differentiation will require the noncanonical Wnt signaling. This dynamic pattern of Wnt activity highlights the developmental complexity of the heart and provides important insights to the generation of cardiomyocytes from hPSCs. In addition, Wnt signaling is important for morphological events such as chamber, valve, and septum formation and cardiac looping. 

In nervous system development, Wnt signaling is also a key regulator of various aspects of corticogenesis (for detailed reviews [21,22]). The establishment of the anterior-posterior axis and neural tube closure is dependent on a Wnt gradient. In the future forebrain, Wnt signaling is important for dorsal specification. During embryonic development, the cortical hem in the medial dorsal forebrain serves as a significant source of Wnt signaling to pattern the hippocampus. While Wnt signaling is active during cortical neurogenesis, its function is context-specific, where it can promote proliferation or differentiation of RGCs. In post-mitotic neurons, Wnt signaling can shape axons, dendrites, and synapses as well as provide cues for the establishment of layer-specific identity. After the switch to gliogenesis, Wnt signaling is also important for both astrocyte [23] and oligodendrocyte [24] development.

The multifaceted role of Wnt signaling renders almost any stage during heart and brain development susceptible to variants affecting Wnt signaling. Indeed, *CTNNB1* is a shared risk gene between CHD and NDD, and dysregulation will lead to abnormal development of both heart and brain structures, e.g., *CTNNB1* null mouse embryos lack formation of mesodermal and head structures [25,26]. Intriguingly, *CTNNB1*^+/−^ mice are indistinguishable from wild-type mice [26], indicating that haploinsufficiency phenotypes are not well understood. The ability to model haploinsufficiency will better recapitulate patient genetics which should be applied to both *CTNNB1* and other CHD and NDD genes since DNMs are predominantly heterozygous LOF mutations [27]. 

### 3.2. How Epigenetic Regulation Shapes the Development of the Brain and Heart

Modification of DNA and histones regulate chromatin structure dynamics and allows for the binding of transcription and related factors to modulate gene expression. Our GO analysis highlights the importance of histone methylation associated with both CHD and NDD. It is widely accepted that lineage-specific genes will be repressed until the cell-fate is determined. The identification of bivalent histone domains in pluripotent stem cells support the concept that promoters and enhancers remain in a poised state with both active (H3K4me3 and H3K4me1/2, respectively) and repressive methylation marks (H3K9me3 and H3K27me3) at the pluripotency stage [28]. As differentiation commences, the bivalent status will resolve to a monovalent status with either gene activation or repression to inhibit alternative fates. Examples of these mechanisms have been demonstrated for both the cardiac and neural lineages where cell-type-specific transcription factors will gradually lose H3K27me3 and gain H3K4me3 marks on the regions to achieve lineage specification [29,30]. It is interesting to note that during differentiation, enhancer regions will acquire H3K4me1 to poise additional genes for activation at a later stage [31,32]. However, expansion of the repressive domain is more prominent during differentiation, suggesting that more genes are repressed than activated as cells exit pluripotency [33]. Besides histone marks in regulatory regions, H3K36me3, which marks active transcription, is also important as it is linked with several key cellular processes such as alternative splicing and DNA repair [34]. Another prominent feature for H3K36 methylation is that there are multiple enzymes that are playing non-redundant roles. This complex nature could partially explain why H3K36 methylation is highly enriched in CHD genes and NDD genes. 

Disruption of chromatin-modifying enzymes such as histone methyltransferases and demethylases are detrimental for accurate developmental programming. Due to the expression of these genes early in development, pathogenic variants are likely to manifest both CHD and NDD (Table 1). The trithorax group proteins (TRX) and polycomb group proteins (PRC, including PRC1 and PRC2) are activating H3K4 and repressive H3K27 methyltransferases, respectively. Several genes associated with these two groups are shared between patients with CHD and NDD. Knockout studies of these factors demonstrate defective heart and brain development (Table 1). Interestingly, while genes related to H3K4 methylation are frequently shared, H3K27 methylation-related genes are not enriched with exception to its demethylase KDM6B. This finding, however, is not in line with current studies that the conditional knockout of EZH2 or EED in cardiac or neural lineages will lead to abnormal heart and neurodevelopment [35,36,37]. One possibility is that current datasets did not completely capture all genetic variants, as patients with pathogenic variants in PRC2 complex proteins (EED, EZH2, SUZ12) are diagnosed with overgrowth syndrome displaying intellectual disability and CHD [38,39,40,41]. However, the degree of intellectual disability is mild-to-moderate with rare co-occurrences of other neurodevelopmental defects such as autism and infrequently demonstrate CHD, which may explain why large cohort studies do not include or are not able to bring these variants to significance. While current studies appear to dampen the role of H3K27me3 in the pathogenesis of CHD and NDD, it still serves as a potential convergent pathway for many of the risk genes, as it is a major repressive mechanism that counteracts gene activation (Table 1). We cannot, therefore, exclude a potentially significant contribution of PRCs to both CHD and NDD.

## 4. Potentially Modeling the Co-Occurrence of CHD and NDD Using Human Pluripotent Stem Cells (hPSCs)

Animal models are instrumental in delineating the correct spatial and temporal timing during development and genetic tools are a powerful resource to unravel the key players in certain developmental programs. The mouse brain has a similar cortical architecture and neural circuits to humans and the developing murine heart shares comparable morphological development with the human fetus [77]. Studying heart development in zebrafish is also attractive due to their unique regenerative ability. While these models help define key pathways and regulators of development, an important caveat is that these models may not fully recapitulate the unique features of human disease, which results in poor efficacy in preclinical models [78]. Primate models, which are genetically and structurally more similar with human, have been used to model *MeCP2* and *SHANK3* mutations in autism [79], as well as adult heart disease such as myocardial infarction [80]. However, the cost and availability of primates are generally intractable and are a barrier to systematically studying the mechanisms underlying pathogenesis. These challenges warrant an accessible human model to investigate early development.

hPSCs represent a powerful source of cells for human disease modeling given their ability to generate a nearly unlimited supply of potentially any cell type of the body. Access to disease-relevant cell types carrying the exact patient-specific genetic information by directed differentiation of hPSCs has yielded an attractive paradigm to study disease onset and progression in vitro. Over the years, various methods have been developed for directed differentiation that is either monolayer- or embryoid body (EB)-based. The hPSC approach has already been applied to generate several human disease models. Below we will summarize advances in producing cell types from hPSCs related to the development of CHD and NDD (Figure 2, Table 2 and Table 3). As a comparison, we will also talk about trans-differentiation methods that skip the hPSC stage to get disease-relevant cell types.

### 4.1. Directed Differentiation of hPSCs to Cardiac Cell Types

#### 4.1.1. 2D Differentiation of hPSCs towards Cardiomyocytes

Highly efficient and reproducible differentiation platforms to generate cardiomyocytes have been established that eliminate the need of xenogeneic material (feeder cells and serum) to promote cell fate specification [110]. These chemically defined methods modulate signaling pathways to promote cardiogenic fate based on prior knowledge of heart development. Treatment of hPSCs with Activin/Nodal/TGF-β and BMP signaling had to be optimized to induce cardiogenic mesoderm formation for different hPSC lines [81,82,83]. A more consistent approach is to utilize the biphasic role of Wnt signaling during mesoderm induction and cardiac specification as mentioned above. The addition of high concentrations of Wnt activator (e.g., CHIR99021 [84,85]) subsequently followed by Wnt inhibition (e.g., IWR-1 [111], IWP-2/4 [84,85]) has been shown to improve the generation of cardiac progenitors. Once the cardiac progenitors are specified, activating Wnt signaling by supplementing factors such as Wnt3a will expand the progenitor pool [112]. As differentiation strategies improve, the use of chemically defined monolayer methods can achieve efficiencies reaching over 80% [85,86]. 

Cardiac progenitors generated from hPSCs will spontaneously differentiate into cardiomyocytes. These cardiomyocytes will comprise sarcomeric ultrastructural patterns [113], produce action potentials, and demonstrate contractility [114]. Transplantation into nonhuman primates with myocardial infarction resulted in the regeneration of hearts highlighting the ability to integrate within an in vivo counterpart [115]. hPSC-derived cardiomyocytes retain an embryonic identity and are vastly immature compared to adult-typed cardiomyocytes, much like what is observed in several other hPSC derivatives. Single-cell transcriptomic analysis of cardiomyocytes in long-term culture (day 30 of in vitro culture) are more immature than cardiomyocytes found in the first trimester of human fetal development [116]. The addition of triiodothyronine (T3) in culture showed an accelerated maturation resembling the cardiomyocytes of a second trimester heart by gene expression profiling [117]. Other methods, such as in vivo transplantation and the application of biophysical and metabolic interventions, are being explored to generate more mature cardiomyocytes [118]. 

To precisely study CHDs, chamber-specific cardiomyocytes are needed as atrial and ventricular cardiomyocytes exhibit different electrophysiological and contractile properties, and some CHDs, such as HLHS, are caused by chamber-specific defects [119]. Without any atrial or ventricular specification cues, hPSC-derived cardiomyocytes will reflect an atrial identity early in differentiation and a ventricular identity late [120]. Retinoic acid (RA) signaling has been shown to play a key role in the specification of the chamber as ventricular cardiomyocytes emerge when RA is inhibited, and atrial cardiomyocytes are found when RA signaling is active [88]. An additional study has shown that distinct mesoderm populations in vitro can give rise to either atrial or ventricular cardiomyocytes [89]. This chamber-specific mesodermal specification is modulated by gradients of BMP and Activin A signaling. Moreover, Wnt and Neuregulin pathways will further specify ventricular cardiomyocytes to trabecular or compact subtypes [90]. The methods described here highlight the ability of hPSCs to generate a wide diversity of cardiomyocyte subtypes for modeling and drug screening applications.

#### 4.1.2. Induced Cardiomyocytes and Trans-Differentiation

Alternative methods have been developed to directly derive cardiomyocytes from somatic cells aiming to preserve the age of the donor cell. Overexpression of cardiac lineage transcriptional factors including Gata4, Mef2c, and Tbx5 (GMT) can reprogram mouse fibroblasts to cardiomyocytes [121]. However, overexpression of GMT is insufficient for the human fibroblast trans-differentiation. Additional factors, in combination with GMT, can promote this conversion, such as Mesp1 and Myocd [122]. 

As most CHD is detectable in the first trimester [123], disease-related mechanisms may have already been established during the progenitor stage before the formation of cardiomyocytes, which may not be captured by the trans-differentiation methods mentioned above. Interestingly, a novel platform was developed aiming to generate cardiomyocytes from fibroblasts through an intermediate cardiac progenitor stage [124]. By transient activation of the four induced pluripotent stem cell reprogramming factors (i.e., Oct4, Sox2, cMyc, and Klf4) in mouse fibroblasts, one can redirect the pluripotent status towards that of a cardiac identity. These induced cardiac progenitor cells will remain proliferative before being induced towards cardiomyocytes; thus, they are less limited by the efficiency of fibroblast conversion.

#### 4.1.3. 2D Differentiation of hPSCs towards Endocardial Cells

Endocardial cells are specialized endothelial cells that line the inner layer of the heart. Therefore, one way to derive these cells from hPSCs is to sort out the endocardial population from hPSC-derived endothelial cells [91]. However, as endocardial cells develop from the cardiogenic mesoderm instead of the hemogenic mesoderm that gives rise to vascular endothelial cells, a more faithful way to generate endocardial cells is to generate cardiogenic mesodermal cells and induce an endothelial fate using factors such as VEGF [87,92]. A more detailed analysis of developmental signaling pathways reveals the role of BMP10 in the specification and maintenance of endocardial cells [93]. Based on marker expression, these hPSC-derived endocardial cells are able to undergo EndoMT to give rise to valvular interstitial cells and promote the trabecular fate of cardiomyocytes, recapitulating the role of endocardial cells in vivo.

#### 4.1.4. Generation of Cardiac Organoids from hPSCs

The recent development of strategies to derive cardiac organoids from hPSCs has provided a promising avenue to investigate cell–cell interactions in a more complex environment. Since the directed differentiation of hPSCs results in early embryonic tissue, multiple groups have taken advantage of this system to successfully generate cardiac or heart forming organoids (HFOs). Precardiac organoids follow a developmental trajectory by sequentially activating and repressing WNT signaling in the presence of BMP to generate FHF and SHF-like structures [96]. Similarly, modulating WNT signaling in combination with embedding the organoids within an extracellular matrix such as Matrigel generates HFOs [97]. These HFOs have an anterior foregut endoderm core surrounded by endocardial cells, a dense layer of cardiomyocytes and a mixture of cardiomyocytes with posterior foregut endodermal cells. Cardiomyocytes from these organoids reflect a SHF-like identity. Another approach applies similar Wnt modulation early and cultures the organoids in a medium that supports multiple lineages [98]. This process generates organoids with a cardiomyocyte core surrounded by endodermal and epicardial cells. While these new models have several cellular components that mimic heart development, the complex morphogenesis of various structures remains absent. Recently, a method using gastruloids in combination with cardiogenic factors recapitulated early heart morphogenesis from heart field and crescent formation to the linear heart tube formation [99]. This study was based on methods developed for the differentiation of mouse PSCs; it will be interesting to see whether cardiogenesis can be recapitulated in human gastruloids [125]. A key hallmark of heart development is chamber formation. Using the Wnt modulation strategy with the addition of BMP4 and Activin A during mesoderm induction promotes spontaneous chamber cavity-like formation before cardiomyocyte specification [100,101]. Four-chambered cardiac organoids have also been generated recently using laminin-entactin complex and FGF4 using mouse ESCs [102].

While the heart consists of several distinct cell types, the Wnt modulation strategy is likely to generate endoderm-derived cell types as Wnt activation is also needed for endoderm specification [126]. This is in line with protocols mentioned above that are able to generate endodermal cells [97,98]. Endocardial and epicardial cells could also spontaneously form and line the chamber cavity [100,101] or outer surface [98,101], respectively. The co-culture of epicardial cells with cardiac chamber organoids was also developed to better mimic the three-layer formation in in vivo heart tissue [100]. Fibroblasts and endothelial cells are similarly found in cardiac organoids, as they are of the same mesoderm origin. 

#### 4.1.5. 2D and 3D Methods to Model CHD

A plethora of CHDs have been modeled using hPSC-based stem cell models, including congenital arrhythmias, cardiomyopathy, valvular, and septal malformations and chamber-specific diseases. Efforts to study the pleiotropic effects of Friedreich’s ataxia combine both neurons and cardiomyocytes to understand disease pathogenesis [127]. Most studies focus on disease mechanisms related to hPSC-derived cardiomyocytes. Some key phenotypes associated with cardiomyocyte pathogenesis can be identified by examining reduced contractility, disorganized sarcomere, abnormal electrophysiology, altered calcium handling, and defective metabolic activity. Other heart-related hPSC models are also utilized. For example, while cardiomyocytes that are differentiated from hiPSCs in HLHS patients show abnormalities in structure and function [128], these could not explain other abnormalities in the valve and septum. Therefore, the derivation of hiPSC-derived endocardial cells in HLHS patients demonstrates that these cells could not support cardiomyocyte growth and have reduced EndoMT transition to form valves [91]. These studies highlight the importance of more complex modeling of CHD using additional cardiac-related cell types to better identify the mechanisms that lead to pathogenesis. While current cardiac organoid technologies are still immature, it will be exciting to investigate their potential for accurately modeling CHD as the technology advances to develop more complex cardiac organoids.

### 4.2. Directed Differentiation of hPSCs to Neural Cell Types

#### 4.2.1. 2D Differentiation of hPSCs towards Neural Progenitors and Their Derivatives

Recent advances in using entirely chemically defined differentiation platforms reduce batch-to-batch variability and efficiently generate neural progenitors with dorsal forebrain identity [94]. The widely used “dual-SMAD” inhibition (dSMADi) protocol, in which the TGF-β and BMP signaling pathways are inhibited, generates a highly pure population (>80%) of dorsal-like neural progenitors in 7–10 days post-induction. This method is able to recapitulate the dynamic developmental signaling during corticogenesis as activation of SHH can pattern neural progenitors towards ventral fates [129], and WNT can pattern towards more caudal fates. The addition of FGF8 after initial patterning using dSMADi plus a Wnt inhibitor will direct neural progenitors towards a prefrontal cortical fate [95], which is thought to be a critical region for NDD pathogenesis [130]. 

hPSC-derived neural progenitors can be maintained and expanded using classic mitogens such as EGF and FGF2. During development, these cells will adopt different subtype identities and give rise to neurons, astrocytes, and oligodendrocytes. Similarly, deep- and upper-layer neurons emerge from hPSC-derived neural progenitors in a time dependent manner mimicking in vivo inside-out pattern of neurogenesis [131]. The cell cycle exit of these progenitors will initiate neuronal differentiation, e.g., by adding γ-secretase inhibitor to inhibit NOTCH signaling [132]. hPSC-derived dorsal forebrain neural progenitors will differentiate towards cortical excitatory neurons. Although neuronal morphology rapidly forms in culture (~5–6 days), it takes several months to achieve functional maturation in terms of electrophysiological profiles and the synaptic transmission reflecting the protracted maturation of the human cortex. After neurogenesis, hPSC-derived neural progenitors become competent to generate glia and enter a phase of gliogenesis [133]. Induction of these glial progenitors towards astrocytes is enriched by factors such as IL-6 family cytokines (CNTF, LIF or CT-1) or BMP that has previously been shown to be important for astrocytic differentiation and specification. Factors that promote the maturation of immature astrocytes are still unknown; however, maintaining cortical spheroids with astrocytes for up to 590 days in culture demonstrates that this process may also be partially a time-dependent mechanism [134]. Strategies to accelerate cortical gliogenesis rely mainly on serum factors and clonal expansion of gliogenic-favored hPSC lines. An alternative strategy is to overexpress factors such as NFIA, NFIB, and SOX9 and in various combinations in hPSCs, neural progenitors, or fibroblasts [135,136,137,138]. The derived astrocytes are functional, as they are able to support hPSC-derived neuron maturation, elicit calcium transients, and become reactive astrocytes upon cytokine treatment. These rapid approaches enable derivation of astrocytes at a similar speed compared with that of neurons.

#### 4.2.2. Induced Neurons and Trans-Differentiation

In addition to directed differentiation methods, the direct induction of neurons from somatic cells has also been achieved. While the expression of neural lineage-specific transcriptional factors Brn2, Ascl1, and Myt1l (BAM) is capable of converting murine fibroblasts into induced neurons [139], human fibroblasts require an additional factor, NeuroD1 [140]. While most induced neurons generated are cortical excitatory neurons, the derivation of induced inhibitory neurons can be achieved through the overexpression of a five-transcription factor cocktail (Foxg1, Sox2, Ascl1, Dlx5, and Lhx6) [141]. Inducing neural progenitors from somatic cells has also been achieved by using a brief pulse of induced pluripotency factors cultured in medium with EGF and FGF2/4 to support neural progenitor expansion [142]. As several NDD genes are expressed early in neurodevelopment in the progenitor stage, generating disease-related neural progenitor directly from somatic cells will be a useful alternative strategy to study how pathogenic progenitors impact neurodevelopment. In addition to inducing neurons from somatic cells, the one-step neuronal induction method was also developed for hPSCs. Overexpressing Ngn2 in hPSCs will induce cortical excitatory neuron in less than seven days [143], which enables rapid phenotype examination if pathogenic variants are expressed exclusively in neurons that will impact neuronal function. 

#### 4.2.3. Generation of Brain Organoids from hPSCs

The development of cerebral organoids is an exciting model to investigate the spatial complexity of the developing brain. The serum-free culture of embryoid body-like quick aggregation (SFEBq) method laid the foundation for deriving cortical spheroids in vitro [103]. These SFEBq spheroids generate neural progenitors with dorsal forebrain identity, organized in a rosette-like pattern, and will give rise to functional and transplantable layer-specific cortical neurons. The addition of an extracellular matrix (e.g., Matrigel) improves the robust growth of the cortical plate from the SFEBq spheroids [104]. Building off this work and improving long term culture conditions using spinning bioreactors resulted in the first report on the development of cerebral organoids [105]. The organoids generated by this method exhibit a rudimentary cortical inside-out pattern, display an oSVZ-like progenitor area, and morphologically mimic fetal brain development up to the second trimester (~24 weeks). Most methods to date are developed based on the initial SFEBq methodology by modulating developmental signals to pattern different areas of the brain. These guided methods differ from the unguided method [105], where no developmental cues are added to the culturing system resulting in the generation of forebrain, midbrain, and hindbrain progenitors. 

Guided methods to generate dorsal forebrain progenitors use dual-SMAD inhibition strategies [106]. Expansion of the progenitor pool with mitogens EGF and FGF2 as well as embedding in Matrigel can improve growth and structural expansion. As the progenitors are fated, the long-term culturing conditions are more focused on supporting neuronal growth and differentiation with the inclusion of neurotrophic factors rather than patterning [107]. Culturing organoids with the extracellular matrix can help the radial alignment of newborn neurons as it provides outside support for the neuronal migration along the radial fibers [144]. Neuronal maturation, including spontaneous firing activity and synaptic transmission, could be detected [105,145]. However, as the organoids grow and mature, a necrotic core at the center of these organoids will develop resulting in the progenitor zone being gradually depleted and reducing the cortical plate formation. The limitation of long-term culture may also explain why cortical organoids stall at the second trimester-like development. To reduce the necrotic core, strategies have been developed to introduce vascular elements, such as transplantation to the mouse brain [146], co-culture with endothelium [147,148], and ectopic expression of hemangiogenic factors [149]. Slice cultures of organoids can also be applied to enhance cellular survival inside and sustain organoid growth and neuronal maturation [150,151].

The functional assessment of the cortical organoids will be integral to identifying NDD related phenotypes. A minimal cortical circuit is comprised of both excitatory and inhibitory neurons and glia. A potential strategy to model these interactions is to derive both dorsal and ventral forebrain organoids and co-culture these together to form assembloids [152,153,154]. Live imaging experiments demonstrate that interneurons derived in the ventral domain are able to migrate tangentially to incorporate in the dorsal domain. However, the timing on when to merge the organoids could potentially affect the efficiency of migration. An innovative approach is to direct the generation of both domains within a single organoid by placing signaling organizers during differentiation. One such study developed a signaling center that secreted SHH, mimicking the notochord during development, and established dorsal and ventral domains in a single organoid, and some interneurons exhibit migrating morphologies [155].

#### 4.2.4. 2D and 3D Methods to Model NDD

hPSC-derived brain models have been widely utilized to model several diseases ranging from NDDs [156] to neurodegenerative disorders [157] as well as assessing brain development and function during infection by ZIKA virus or SARS-CoV-2 [158]. Modeling NDD mainly focuses on the early program of neurogenesis. A pioneering study utilized forebrain organoids to demonstrate that FOXG1 overexpression will increase the formation of interneurons, leading to an excitatory-inhibitory neuron (E/I) imbalance that can reflect disease pathogenesis described in autism and other NDDs [109]. The generation of regionally restricted assembloids can provide insight to the interactions between different regions of the brain. For example, recent studies utilizing forebrain assembloids identified E/I imbalance mechanisms in Timothy syndrome [152]. Electrophysiology of hPSC-derived neurons is important for assessing how neuronal functions are disturbed in NDDs including changes in spontaneous neuronal activity, network activity, synaptic transmission, and calcium signaling [159,160]. In addition to the neuronal function, how mutations in NDD risk genes affect cortical neural progenitor proliferation and differentiation have been examined for several genes [150,161,162,163]. These above-mentioned advances provide a robust platform to model NDDs and the diversity of tools used to analyze the impacts of NDD risk genes on neurodevelopment. 

### 4.3. Comparison among 2D, 3D and Trans-Differentiation Methods

As multiple strategies exist, choosing a proper model will be crucial for answering research questions. Trans-differentiation is the fastest way to acquire disease-relevant cell types to examine phenotypes, compared with time-consuming 2D or 3D differentiation. However, trans-differentiation skips all essential steps during development and, therefore, will only be able to detect defects in the target cell type but not the whole developmental process. In addition, compared with directed differentiation, efficiency for trans-differentiation is low and unable to generate different layer-specific neurons or chamber-specific cardiomyocytes. Furthermore, overexpression of key lineage factors does not occur in normal development, therefore, it may mask any potential phenotypes that are associated with patients. 

As some CHD and NDD appear to alter normal development, a more reliable way to recapitulate the pathogenesis is to utilize 2D or 3D differentiation from hPSCs and examine any defects throughout. Using 2D methods is suitable for various cell-based assays as they generate a large quantity of homogenous disease-relevant cells. The use of 3D methods is suitable for investigating cell–cell interaction and examining any structural defects that disrupt normal self-organization. However, these 2D and 3D methods are amenable to batch-to-batch difference and variability across different hPSC lines. Therefore, consistent results from repeated experiments will be more reliable and convincing to infer the potential pathogenesis.

## 5. Screening Strategies for Identifying Phenotypes Caused by Shared Genetic Variants

While stem cell models provide us with a powerful tool to study human development and pathology, a challenge is to investigate how shared genetic variants in CHD and NDD may confer risk for disease. To determine whether these variants give rise to measurable in vitro phenotypes, rapid screening approaches are ideal. Cell line pooling approaches are attractive because a single differentiation may uncover cellular phenotypes from several variants at once. Two technical obstacles must be resolved in order to achieve this: how do we acquire cells with different genotypes in a large-scale manner, and what will be the ideal phenotype to measure that will be altered by genotypes (Figure 3).

Several strategies exist and have been widely used to acquire genotypes of interest. Genome-wide CRISPR screens have been developed to investigate the role of particular genes in several biological processes. Pooled shRNA and CRISPR interference screens also serve as powerful methods to reduce target gene expression and can provide insights into the function of essential genes without introducing mutations [164,165]. However, in several cases of CHD and NDD, disease-relevant genetic variants result in haploinsufficiency or are missense variants; thus, rapidly engineering patient-like lines using CRISPR/Cas9 remains technically challenging. Advanced editing methods such as base [166] and prime editing [167] facilitate the generation of patient-specific genetic variants, and future advances in these technologies can potentially drive the rapid generation of multiple mutations. Alternatively, large-scale collections of hiPSCs with patient-specific genetic variants can serve as another resource to study the impact of variants in pathologic processes [168,169]. By pooling several patient hiPSC lines together, one can investigate the impact of multiple variants within a dish. This strategy can reduce well-to-well variability and save on media and growth factor costs. However, differences in the genetic background of pooled iPSCs may introduce variations in the molecular readouts that are not directly caused by the pathogenic variant [170,171]. Therefore, multiple hiPSCs with different genetic backgrounds but similar genetic variants should be considered to design the pooled hiPSCs study [172].

To identify molecular and cellular phenotypes in a population, cell survival or proliferation assays are often utilized as a primary screen to categorize genotypes [173]. For more directed assays, target gene expression via fluorescence reporters or cell-surface markers can be combined with flow cytometry to isolate populations that deviate from wild-type controls. Electrophysiological- or calcium-signaling assays will be able to pinpoint functional consequences in mutant neurons or cardiomyocytes [174,175]. Combining functional assays with mRNA sequencing such as Patch-seq [176] will directly link functional readouts with gene expression. Perturb-seq [177] and CROP-seq [178] methods can determine the impacts of genetic variation within complex cell populations such as organoids using single-cell RNA seq (scRNA-seq). This is particularly powerful to identify lineage susceptibilities caused by different genetic variants. Recent work has applied Perturb-seq on in vivo mouse brains to investigate how NDD risk genes affect cortical development [179], and a similar study in endoderm patterning has been applied using iPSCs [168]. Compared with directed assays, transcriptomic profiling during genetic screens will also help discover convergent molecular mechanisms among different genetic variants. Global gene expression alterations or changes of alternative splicing based on RNA-seq will survey common signatures to disentangle genetic complexity and develop common druggable target. As single-cell multi-omics (transcriptome, epigenome, and proteome) is emerging as a conventional assay, pooled screening methods will provide novel insights into the convergent mechanisms in multiple cell types to quickly unravel disease associated mechanisms.

## 6. Therapeutic Potential for hPSC-Based Models to Treat CHD and NDD

Understanding the underlying mechanisms caused by genetic variants can pave the way for personalized treatment (Figure 3). hPSCs are highly scalable and enable rapid establishment of a high-throughput drug-screening platform. Readouts are similar with those that are used in genetic screening, including survivability, fluorescence imaging, and functional indicators. The screening assay will depend on the characterized phenotype. Examples of these efforts include studies on the identification of drugs that inhibit ZIKV infection or ZIKV-induced neural progenitor death [180,181], or drugs that will suppress calcium-handling abnormalities in arrhythmogenic hiPSC-derived cardiomyocytes [182]. Recent technical advances have made it possible to examine cellular architecture recovery in mutant organoids by drugs in a high-throughput manner. For example, the tissue clearing method of organoids obviates the need for intensive tissue sections and enables high-content imaging analysis of different cell components inside organoids. This strategy has been applied to the drug screening by brain organoids from patients with Alzheimer’s disease [183]. Additionally, scRNA-seq could also be applied to the drug screening with each drug-treated organoid with one unique barcode [184].

Cell replacement therapy is another avenue to pursue regarding personalized treatment. Cell replacement therapies in CHD focus on somatic stem cells to treat a limited set of diseases, including HLHS and cardiomyopathy [185]. Although encouraging results have been reported, mechanisms on how these stem cells work to regenerate are thought to be due to paracrine effects, not really replacement by differentiated cardiomyocytes from administered cells in vivo. The use of hPSC-derived cell types could overcome challenges facing traditional therapies. Patient-derived hiPSC could first be corrected for genetic mutation. Different cardiac cell types, including different types of cardiomyocytes and endocardial cells, could then be generated from hPSCs under defined conditions, which will not only avoid direct transplantation of stem cells but also allow for cell-type-specific restoration that has the potential to treat a wider range of diseases such as valvular diseases and arrhythmia. While replacement by hPSC-derived terminally differentiated cardiac cells is prospective for CHD, replacement by hPSC-derived neurons for NDD is not as promising. This is mainly because functional brain defects, but not neuronal loss, are the primary mechanism for NDD compared with neurodegenerative disorders. As neuronal networks start to be built from the very beginning of neurodevelopment, replacement by hPSC-derived neural progenitors will be more reasonable. In addition, these hPSC-derived cells provide the opportunity for the autologous transplantation of neural progenitors and are more suitable to treat NDD as they exhibit an early neurodevelopmental phenotype according to current protocols.

## 7. Conclusions

The co-occurrence of CHD and NDD is largely believed to be due to non-genetic factors leading to cortical malformation [186], however, massive sequencing efforts have identified hundreds of risk genes in patients with CHD and/or NDD for which many appear shared. The differentiation of hPSCs into disease-relevant cells and tissue is an attractive platform to investigate the role of shared genetic variants in an early window of human development. Future challenges for these models include developing strategies to transition from fetal-like to a postnatal-like identity by improving long-term cultures, accelerating the maturation of the cardiac and cortical networks within an organoid, and improving organ-like structural components. It will be exciting to see what the next decade of disease modeling will bring as more technical hurdles will be overcome. Bioengineering approaches will facilitate this advancement by developing microfluidic chips and automated systems to more precisely control the development of tissue organoids.

## Figures and Tables

**Figure 1 cells-11-00460-f001:**
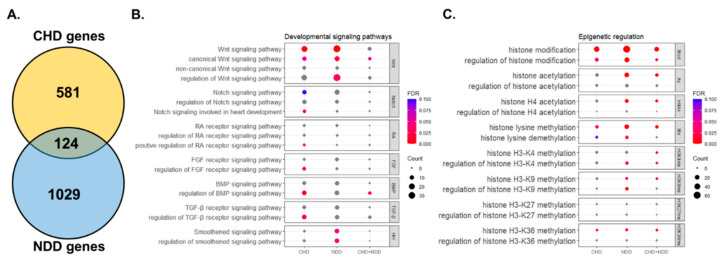
Gene Ontology (GO) analysis of shared genetics in both CHD and NDD. (**A**). Venn diagram representing overlap between high-risk genes in CHD [6] and NDD [5] used for GO analysis (GO biological process). (**B**). Dot plot graph (number of input genes in each GO term) and significance (false discovery rate, FDR) in GO terms related with developmental signaling pathways. (**C**). Similar to B, but with GO terms related with epigenetic regulation.

**Figure 2 cells-11-00460-f002:**
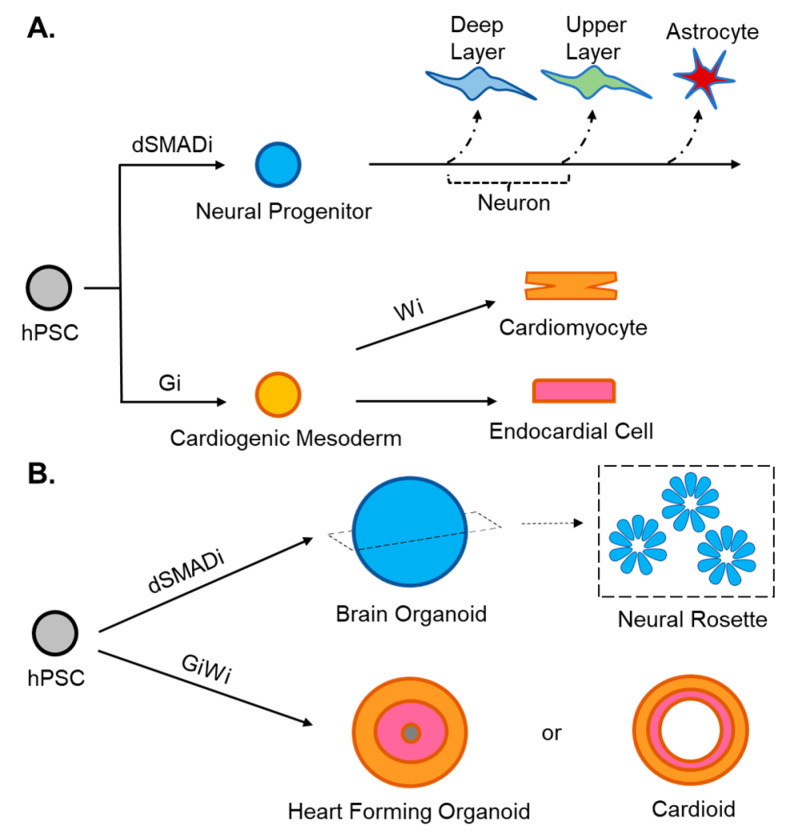
Diagrams providing overview of 2D and 3D differentiation methods. (**A**). Overview of 2D differentiation towards cortical neural progenitors and cardiogenic mesoderm. Neural progenitors can be generated using the dual SMAD inhibition (“dSMADi”) protocol with high efficiency. Cardiogenic mesoderm cells are efficiently derived by activating Wnt signaling (e.g., GSK-3β Inhibitor CHIR-99021, termed “Gi”). They can be further differentiated towards cardiomyocytes by inhibiting Wnt signaling (“Wi”), or towards endocardial cells. (**B**). Small molecules methods used in 2D differentiation can also be incorporated into 3D differentiation methods. However, levels and duration of factors will likely need to be reoptimized.

**Figure 3 cells-11-00460-f003:**
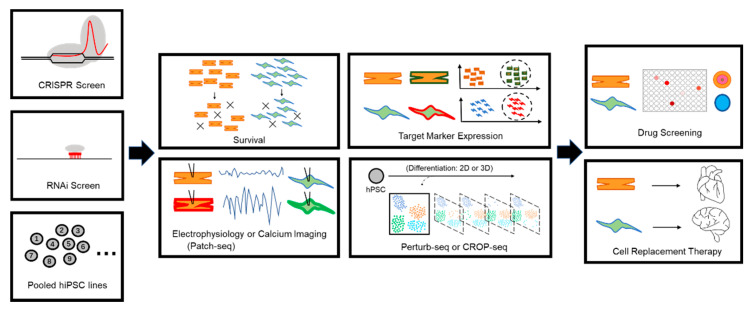
Schematic for potential workflows to investigate shared genetic variants in CHD and NDD.

**Table 1 cells-11-00460-t001:** Summary of CHD and NDD genes that are related with major histone modification. Genes that are related with PRC complex are marked bold: KMT2A, KMT2C, KMT2D, and ASH1L are trithorax group proteins that counteract with polycomb group proteins [42]. KDM5B affect binding of polycomb group proteins in a subset of genes [43]. CHD7-bound regions are depleted of H3K27me3 [44,45]. KDM6B removes H3K27me3 marks [46]. NSD1 has been shown to modulate PRC2 activity [47,48]. WHSC1 has been shown to alter binding of EZH2 [49]. References that talk about heart or brain phenotype in model systems caused by mutations of corresponding gene are marked behind each gene.

Group	CHD and NDD Genes That are Related with Major Histone Modification
H3-K4	Methyltransferase: **KMT2A** (brain [50], heart), **KMT2C** (brain [51,52], heart [53]), **KMT2D** (brain [54,55], heart [56]) Demethylase: **KDM5B** (brain [43,57], heart) Methyl group reader: **CHD7** (brain [58], heart [59])
H3-K9	Methyl group reader: POGZ (brain [60,61], heart) Acetyltransferase: KAT6A (brain, heart [62,63]), KAT6B (brain [63,64], heart)
H3-K27	Demethylase: **KDM6B** (brain [65], heart [66]) Acetyltransferase: EP300 (brain [67,68,69], heart [70])
H3-K36	Methyltransferase: **NSD1** (brain [63], heart), **WHSC1** (brain [71], heart [71,72]), **ASH1L** (brain [73,74], heart), SETD5 (brain [75], heart [76])

**Table 2 cells-11-00460-t002:** Summary of two dimensional (2D) methods for hPSC-based models. D0 marks the start of differentiation. D0-D1 is referred as the first 24 h, D1-D2 as the next 24 h, and so on. See the corresponding reference for more details.

Cell Type	Base Medium	Differentiation Protocol	EB or Monolayer	Efficiency	References
Cardiomyocyte	RPMI-B27 (d0–)	Activin A (d0–d1) BMP4 (d1–d5) *D12: widespread contracting activities*	Monolayer	>30% >70% (with percoll gradient enrichment)	[81]
Cardiomyocyte	StemPro-34 based medium (d0–)	BMP4 (d0–d1) BMP4 + bFGF + Activin A (d1–d4) VEGF + DKK1 (d4–d8) VEGF + DKK1 + bFGF (d8–) *D10: emergence of contracting EBs*	EB	~40% in aggregation culture >50% in monolayer culture (both from KDR^low^/C-KIT^neg^ cells sorted on d6)	[82,83]
Cardiomyocyte	RPMI-B27 minus insulin (d0–d7) RPMI-B27 (d7–)	CHIR99021 (d0–d1) IWP2 or IWP4 (d3–d5) *D12: robust contracting activities*	Monolayer	>80%	[84,85]
Cardiomyocyte	CDM3 (with RPMI, rHA, AA 2-P, d0–)	CHIR99021 (d0–d2) Wnt-C59 (d2–d4) *D7-D9: contraction begins*	Monolayer	>80% >95% (with chemically defined metabolic selection)	[86]
Cardiomyocyte	RPMI-B27 minus insulin (d0-d7) RPMI-B27 (d7-)	CHIR99021 (d-1–d0) Activin A + Matrigel (d0–d1) BMP4 + CHIR99021 (d1–d3) XAV939 (d3–d5) *D7: contraction begins*	Monolayer	~90%	[87]
Chamber-specific cardiomyocyte	RPMI-B27 (d0–)	BMP4 + bFGF (d0–d1) Activin A (d1–d2) Noggin (d3–d5) DKK1 (d5–d11) RA (d5–d8, atrial) RA inhibitor (d5–d8, ventricular)	Monolayer	RA treated: ~50% atrial-like CMs RA inhibitor treated: ~50% ventricular-like CMs	[88]
Chamber-specific cardiomyocyte	StemPro-34 based medium (d0-)	BMP4 (d0–d1) BMP4 + bFGF + Activin A (d1–d3) IWP2 + VEGF (d3–d5) VEGF (d5–d12) ATRA or retinol (d3–d5, atrial)	EB	Higher Activin A to BMP4: ventricular CMs Lower Activin A to BMP4: atrial CMs (variable efficiency across lines)	[89]
Compact or trabecular ventricular cardiomyocytes	StemPro-34 based medium (d0–d10) DMEM high glucose (d10–d16)	BMP4 (d0–d1) BMP4 + bFGF + Activin A (d1–d3) IWP2 + VEGF (d3–d5) VEGF (d5–d10) CHIR99021 + IGF2 + Insulin (d10–d16, compact) NRG1 (d10–d16, trabecular)	EB	Compact: >80% Trabecular: >80%	[90]
Endocardial cells	RPMI-B27 minus insulin (d0–d4) EGM-2 (d4–)	CHIR99021 (high, d0–d2) CHIR99021 (low, d2–d4) VEGF + bFGF (d4–d9) Sorted by CD144+ (d9, endothelium) Sorted by NPR3+ (from CD144+ population, endocardium)	Monolayer	~20% of CD144+ cells are NPR3+	[91]
Endocardial cells	RPMI-B27 minus insulin (d0–d2) StemPro-34 based medium (d2–d5) EGM (d5–)	CHIR99021 (d-1–d0) Activin A + Matrigel (d0–d1) BMP4 + CHIR99021 (d1–d2) BMP4 + bFGF + VEGF (d2–d5) VEGF + bFGF + CHIR99021 (d5–)	Monolayer	>90%	[87]
Endocardial cells	RPMI-B27 with SU5402 (d0–d3) RPMI-B27 (d3–)	Wnt3a (d0–d1) Wnt3a + BMP2 (d1–d2) BMP2 (d2–d3) Sorted by SSEA-1 (d3, cardiogenic mesoderm) VEGF + FGF8 + FGF2 (d3–d9) Sorted by CD31 (d9, endothelial cells)	Monolayer (with feeder)	95%	[92]
Endocardial cells	StemPro-34 based medium (d0–)	BMP4 (d0–d1) BMP4 + bFGF + Activin A (d1–d3) bFGF (d3–d5) bFGF + BMP10 (d5–d9)	EB (d0-d3) Monolayer (d3-)	~50%	[93]
Neural progenitors (dorsal forebrain)	E6 (d0–)	SB431542+ LDN193189 (d0–) XAV939 (d0–d3)	Monolayer	>80%	[94]
Neural progenitors (prefrontal dorsal forebrain)	E6 (d0–d6) N2-B27 (d6–)	SB431542 + LDN193189 (d0–d6) FGF8 + SHH (d6–d20) FGF8 (d20–d40)	Monolayer	>90%	[95]
Neurons (prefrontal dorsal forebrain)	Neurobasal-B27 (d0–)	*Start from Neural progenitors (dorsal forebrain)* DAPT (d0–d5) BDNF + GDNF + AA (d6–)	Monolayer	>90%	[94]

**Table 3 cells-11-00460-t003:** Summary of three dimensional (3D) methods for hPSC-based models. D0 marks the start of differentiation. D0-D1 is referred as the first 24 h, D1-D2 as the next 24 h, and so on. See the corresponding reference for more details.

Type	Base Medium	Differentiation Strategies	Components	Morphological Development	Disease Modeling	References
Precardiac organoid	RPMI-B27 minus insulin (d0–)	Spheroid formation (d0–d2) BMP4 + CHIR99021 (d0–d2) XAV939 (d3–d5)	CXCR4+: *SHF cells* CXCR4−: *FHF cells*	No	N/A	[96]
Heart-forming organoid	RPMI-B27 minus insulin (d0–d7) RPMI-B27 (d7–)	Spheroid formation (d-4–d-3) Matrigel embed (d-2) CHIR99021 (d0–d1) IWP2 (d3–d5)	Cardiomyocytes: *more of SHF identity* Endocardial cells Endothelial cells Foregut endoderm	No	*Nkx2.5*-Knockout	[97]
Multilineage organoid with heart and gut	RPMI-B27 minus insulin (d0–d7) Epicardial permissive medium (DMEM/F12 based, with AA, d7-)	CHIR99021 (d0–d1) IWP2 (d3–d5) Spheroid formation (d5–d6)	Cardiomyocytes: ventricular, atrial, nodal Endocardial cells Epicardial cells Mid-hindgut endoderm	No	N/A	[98]
Gastruloid with cardiogenesis (mouse)	N2-B27 (d0–)	Gastruloid formation (d0–d2) CHIR99021 (d2–d3) bFGF + VEGF + AA (d4–d6)	All three germ layers Cardiomyocytes: *FHF & SHF* Endocardial cells	SHF & FHF domains establishment Cardiac crescent formation Early heart tube formation	N/A	[99]
Cardioid	CDM (d0–)	Spheroid formation (d-1–d0) FGF2 + LY294002 + Activin A + BMP4 + CHIR99021 (d0–d1.5) BMP4 + FGF2 + IWP2 + Insulin + RA (d1.5–d5.5) BMP4 + FGF2 + Insulin (d5.5–d7.5) Insulin (d7.5–)	*Similar with FHF* Cardiomyocytes Endothelial cells	Chamber formation	N/A	[100]
Human heart organoid	RPMI-B27, minus insulin (d0–d6) RPMI-B27 (d6–)	Spheroid formation (d-2–d0) CHIR99021 + BMP4 + Activin A (d0–d1) Wnt-C59 (d2–d4) CHIR99021 (d7, for 1 h)	*Exhibit features of both FHF and SHF* Cardiomyocytes: atrial, ventricular Endocardial cells Endothelial cells Cardiac fibroblasts Epicardial cells	Chamber formation	Pregestational diabetes induced CHD	[101]
Murine heart organoid (mouse)	Heart organoid medium (DMEM/F12 based, d0–)	Spheroid formation (d-4–d0) Transferred onto LN/ET gel (d0) FGF4 (d0–d15, or longer) BMP4 + BIO + LIF (d9–)	Cardiomyocytes: ventricular, atrial, nodal Conducting tissues Smooth muscle cells Endothelial cells	Cardiac crescent formation Heart tube formation Heart tube looping Chamber formation	N/A	[102]
Cortical spheroid	Glasgow-MEM (with additional factors, d0–d18) DMEM/F12 based medium (d18–)	Spheroid formation (d0–d1) IWR1e + SB431542 (d0–d18)	Neural progenitors Neurons	*Only dorsal forebrain* Neural rosette (with VZ, SVZ, oSVZ, IZ) Inside-out pattern of neurogenesis	N/A	[103,104]
Cerebral organoid	Human ES media (with low bFGF, d0–d6) Neural induction media (DMEM-F12 based, d6–d11) Differentiation media (DMEM/F12 & Neurobasal based, d11-)	Spheroid formation (d0–d1) Matrigel embed (d11)	Neural progenitors Neurons	*Forebrain, midbrain, hindbrain* Neural rosette (with VZ, SVZ, oSVZ, IZ) Inside-out pattern of neurogenesis	Microcephaly	[105]
Cortical spheroid	KSR based medium (d0–d6) Neural medium (Neurobasal based, d6–)	Spheroid formation (d0–d1) Dorsomorphin + SB431542 (d0–d6) EGF + FGF2 (d6–d25) BDNF + NT3 (d25–d43)	Neural progenitors Neurons Astrocytes	*Only dorsal forebrain* Neural rosette (with VZ, SVZ, oSVZ, IZ) Inside-out pattern of neurogenesis	N/A	[106]
Cortical organoid	Stem cell medium (DMEM/F12 based, d0–d5) Induction medium (DMEM/F12 based, d5–d13) Differentiation medium (DMEM/F12 based, d13–d70) Maturation medium (Neurobasal based, d70–)	Spheroid formation (d0–d1) Dorsomorphine + A83–01 (d0–d5) Wnt-3A + CHIR99021 + SB431542 (d5–d13) Matrigel embed (d6) BDNF + GDNF + AA + TGF-β + cAMP (d70–)	Neural progenitors Neurons Astrocytes	*Only dorsal forebrain* Neural rosette (with VZ, SVZ, oSVZ, IZ) Inside-out pattern of neurogenesis	ZIKV infection	[107]
Cortical organoid	Induction medium (DMEM/F12 based, d0–d10) Differentiation medium (DMEM/F12 & Neurobasal based, d10-)	Spheroid formation (d0–d1) LDN193189 + SB431542 + XAV939 (d0–d10) BDNF + AA (d18–)	Neural progenitors Neurons	*Only dorsal forebrain* Neural rosette (with VZ, SVZ, oSVZ, IZ) Inside-out pattern of neurogenesis	N/A	[108]
Cortical organoid	Neuronal medium (DMEM-F12 based, d0–d14) Neurobasal-based medium (d14–)	Spheroid formation (d-2–d0) Noggin (d0–d5) Transferred onto Matrigel coated plate (d4) FGF2 + Noggin + Dkk1 (d5–d8) EGF + FGF2 (d8–d14) BDNF + GDNF + AA + cAMP (d14–)	Neural progenitors Neurons	*Only dorsal forebrain* Neural rosette (with VZ, SVZ, oSVZ, IZ) Inside-out pattern of neurogenesis	ASD	[109]

## Data Availability

All the data are presented in this manuscript.

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
