# Peer review of "Harnessing the Power of Stem Cell Models to Study Shared Genetic Variants in Congenital Heart Diseases and Neurodevelopmental Disorders"

_cells, 2022, doi:10.3390/cells11030460_

Round 1
Reviewer 1 Report
In this Review, the authors explore genetic similarities between cardiac and neural development, and expand upon how stem cells could be used explore disease-associated mutations in each of these organs. Some of the genetic links were already known while some arose from their novel comparative analysis between 2 publicly available datasets with de novo mutations in patients with congenital heart disease (CHD) or neurodevelompental disorders (NDD).
The format is a little odd: a VERY brief review of heart and brain development, then novel computational analysis of 2 datasets, followed by separate, independent review of the current state of differentiating hPSCs into cardiac cells (section 7.1) and neurons (section 7.2). Based on the title, I was expecting more details about convergent genetics between cardiac and neural disease, but it was pretty limited.
Additional specific comments below:
- It appears that the authors performed novel analysis by comparing 2 previously published datasets. Normally new data should be published in a research article format rather than a review, but I will let the Editors decide on if this analysis is appropriate.
- The organization of Figure 1 is odd, with the lettering A-I not following a logical flow. Resolution of figure is very poor, can't read the tables. I'd recommend removing the tables from the figure and instead present them in a separate table(s), which is more standard.
-
The authors did a pretty good job summarizing the cardiac and neural differentiation methods. However, it would be more informative to include a table(s) to highlighting targeted signaling pathways, agonists/antagonists used in the differentiation, manipulated TFs or enzyme encoding genes used for the differentiations.
-
I believe a discussion comparing the advantages and disadvantages of different culture techniques (2D, trans-differentiation and 3D cultures) when modeling disease genes would be useful.
- The authors should expand on how discoveries from the iPSC differentiations could provide insights into therapeutic avenues in the future.
Reviewer 2 Report
the authors provided a comprehensive manuscript on stem cell models in congenital heart disease and neurodevelopmental disorders. The manuscript is full of information. It is not easy to read and understood due the huge amount of information the authors decide to share. I suggest to try to reduce the text in order to be not confounding.
I suggest also to include some more images or figures. This may help in quick understanding the message of the authors want to communicate.
The authors also named some tables in the main document. These are not present in the main document but are in supplementary material. I did not understand this choice. Moreover, I did not found any caption in main text on table 1. I suggest to revise the number and the place of tables in main text.
Finally, due to statements in introduction and chapter 2, I guess to find a possible comment on further studies these cells may be involved to understand the common way between CHD and NDD. This may increase the value of the manuscript
Round 2
Reviewer 2 Report
The authors reviewed extensively the manuscript. They replied to the required issues. The final result is an upgraded version.
This version is suitable for publication.